# Association between clinical, serological, functional and radiological findings and ventilatory distribution heterogeneity in patients with rheumatoid arthritis

Elizabeth Jauhar Cardoso Bessa[1], Felipe de Miranda Carbonieri Ribeiro[2,3], Rosana Souza Rodrigues[4,5], Cláudia Henrique da Costa[1], Rogério Rufino[1], Geraldo da Rocha Castelar Pinheiro[1], Agnaldo José Lopes[1]*

1 Postgraduate Programme in Medical Sciences, School of Medical Sciences, State University of Rio de Janeiro, Rio de Janeiro, Brazil, 2 Clinica Felippe Mattoso, Grupo Fleury, Rio de Janeiro, Brazil, 3 Samaritano Hospital, Rio de Janeiro, Brazil, 4 D'Or Institute for Research and Education, Rio de Janeiro, Brazil, 5 Department of Radiology, Federal University of Rio de Janeiro, Rio de Janeiro, Brazil

☯ These authors contributed equally to this work.

* alopes@souunisuam.com.br

**Data Availability Statement:** The data are available at the data repository: https://osf.io/ewj38/.

## Abstract

### Background

In rheumatoid arthritis (RA), the involvement of the pulmonary interstitium can lead to structural changes in the small airways and alveoli, leading to reduced airflow and maldistribution of ventilation. The single-breath nitrogen washout ($SBN_2W$) test is a measure of the ventilatory distribution heterogeneity and evaluates the small airways. This study aimed to find out which clinical, serological, functional and radiological findings are useful to identify RA patients with pathological values of the phase III slope (SIII) measured by the $SBN_2W$ test.

### Methods

This was a cross-sectional study in which RA patients were assessed using the Health Assessment Questionnaire-Disability Index (HAQ-DI) and the Clinical Disease Activity Index (CDAI) and underwent serological analysis of autoantibodies and inflammatory markers. In addition, they underwent pulmonary function tests (including the $SBN_2W$ test) and chest computed tomography (CT).

### Results

Of the 60 RA patients evaluated, 39 (65%) had an SIII >120% of the predicted value. There were significant correlations between SIII and age (r = 0.56, p<0.0001), HAQ-DI (r = 0.34, p = 0.008), forced vital capacity (FVC, r = -0.67, p<0.0001), total lung capacity (TLC) (r = -0.46, p = 0.0002), residual volume/total lung capacity (TLC) (r = 0.44, p = 0.0004), and diffusing capacity of the lungs for carbon monoxide (r = -0.45, p = 0.0003). On CT scans, the subgroup with moderate/severe disease had a significantly higher SIII than the normal/minimal/mild subgroup (662 (267–970) vs. 152 (88–283)% predicted, p = 0.0004). In the final

**Funding:** Initials of the author who received all the awards: AJL Conselho Nacional de Desenvolvimento Científico e Tecnólogico [CNPq; Grant number #301967/2022-9], Brazil. URL: https://www.gov.br/cnpq/pt-br Fundação Carlos Chagas Filho de Amparo à Pesquisa do Estado do Rio de Janeiro [FAPERJ; Grant numbers #E-26/010.002124/2019, #E-26/211.187/2021, #E-26/211.104/2021, and #E-26/200.929/2022], Brazil. URL: https://www.faperj.br/ Coordenação de Aperfeiçoamento de Pessoal de Nível Superior [CAPES, FinanceCode 001, 88881.708719/2022-01, and 88887.708718/2022-00]. URL: https://www.gov.br/capes/pt-br Funders did not play any role in the study design, data collection and analysis, decision to publish, or preparation of the manuscript.

**Competing interests:** The authors have declared that no competing interests exist.

multiple regression model, FVC, extent of moderate/severe involvement and age were associated with SIII, explaining 59% of its variability.

## Conclusions

In patients with RA, FVC, extent of lung involvement and age, all of which are easily obtained variables in clinical practice, identify poorly distributed ventilation. In addition, the presence of respiratory symptoms and deteriorated physical function are closely related to the distribution of ventilation in these patients.

## Introduction

Rheumatoid arthritis (RA) is a systemic disease characterized by articular and extra-articular manifestations and is considered one of the most common immune-mediated diseases worldwide, affecting approximately 1% of the world's population [1, 2]. The main feature of RA is symmetrical inflammatory involvement of the peripheral joints, particularly polyarthralgia and edema of the small joints in the hands and feet. It occurs more frequently in women than in men, at a ratio of 3:1 [3, 4]. Extra-articular manifestations such as subcutaneous nodules and vasculitis are well recognized, although these manifestations can be minimized with more rigorous control of disease activity [5]. These extra-articular manifestations are usually related to a worse prognosis, and pulmonary involvement significantly contributes to morbidity and mortality [3, 6–8]. The most striking feature of lung involvement in RA is the involvement of almost all components of the lung structure, with parenchymal abnormalities including interstitial lung disease (ILD) and nodules, while airway abnormalities include bronchiectasis, constrictive/follicular bronchiolitis and bronchial wall thickening. Some risk factors may be related to the development of RA-ILD, such as smoking, advanced age, high titers of rheumatoid factor (RF) and anti-cyclic citrullinated peptide antibodies (anti-CCP), family history and male sex [9–11].

Pulmonary function tests (PFTs) and chest computed tomography (CT) are the main tests for the initial evaluation and follow-up of patients with pulmonary involvement in RA. However, due to the use of ionizing radiation, chest CT is employed judiciously. Spirometry is an exercise-dependent test that provides a global measure of pulmonary function while being insensitive to regional involvement related to ventilation distribution, small airway disease (SAD) or the early stages of pulmonary involvement. The same is true of chest CT in the detection of peripheral airway obstruction and the early detection of pulmonary fibrosis. In fact, patients with preclinical disease may have abnormal chest CT and normal traditional PFTs [11, 12]. In these patients, involvement of the pulmonary interstitium via inflammation or fibrosis can lead to structural changes in the small airways as well as in the alveoli, resulting in loss of airflow that is reflected in maldistributed ventilation [13]. Techniques for assessing small airway function and ventilation distribution—such as the single-breath nitrogen washout test ($SBN_2W$) and impulse oscillometry—, on the other hand, are very sensitive to the initial changes that occur in lung structure and function and are therefore useful for the early detection of functional abnormalities, even when other tests are normal, or for confirming airflow obstruction when other tests are only subtly abnormal [14–16].

The $SBN_2W$ assesses the ventilatory distribution heterogeneity and can be used as a complementary examination to spirometry in assessing SAD. In recent years, the development of robust accurate commercial devices and the standardization of quality control of washout systems allowed the inert gas washout test to reliably evaluate the ventilatory distribution

inhomogeneity [17, 18]. At the same time, there has been increasing application of the $SBN_2W$ test in a wide variety of clinical conditions, including SAD, asthma, chronic obstructive pulmonary disease (COPD) and systemic sclerosis (SSc) [15, 16, 19–21]. However, there are few studies evaluating ventilation inhomogeneities in RA [22–24] in the early stages of the disease. In fact, the $SBN_2W$ test is able to detect changes in the distribution of ventilation in the small airways when other PFTs are normal [23]. The phase III slope (SIII) is the index derived from the $SBN_2W$ test that represents alveolar gas; specifically, the SIII is the change in $N_2$ concentration between 25–75% of the expired volume during the SBN2W test. Although the $SBN_2W$ test is poorly accessible and is expensive and difficult to perform, it is highly accurate in detecting ventilation maldistribution [21].

Several correlations have been identified between SIII and parameters for monitoring lung diseases. In patients with COPD, SIII was associated with dyspnea, exercise-induced desaturation, lung mechanics and the degree of emphysema [20, 25]. In asthma, ventilation heterogeneity assessed by SIII represents an important indicator of poor control and a high exacerbation rate [15]. In SSc patients, SIII was associated with restrictive damage, changes in pulmonary diffusion and CT patterns [21]. In RA, SIII was associated with RF positivity, bronchial involvement on CT and exercise functional capacity [22–24]. In view of the relative sensitivity of the $SBN_2W$ test and the various associations between SIII and indices used in the assessment of pulmonary and systemic conditions, we hypothesize that findings suggestive of pulmonary involvement in RA used in clinical practice (including the presence of respiratory symptoms, autoantibodies, worsening lung function and appearance of CT abnormalities) may explain the inhomogeneity of ventilation. Thus, the objective of the present study was to find out which clinical, serological, functional and radiological findings are useful to identify RA patients with pathological values of SIII measured by the $SBN_2W$ test.

## Methods

### Patients

This was a cross-sectional study with RA patients aged $\geq$ 18 years of both sexes conducted at the Piquet Carneiro Polyclinic, State University of Rio de Janeiro, Brazil, between August 2018 and February 2022. The study patients were diagnosed with RA by a rheumatologist according to the classification criteria defined by the American College of Rheumatology and the European Alliance of Associations for Rheumatology [26]. The following exclusion criteria were applied: smokers or former smokers with a smoking history >10 pack-years; the presence of other chronic lung diseases and/or sequelae of previous lung diseases unrelated to RA; history of previous pulmonary resection surgery; respiratory infection within the last three weeks; pregnancy; history of COVID-19 pneumonia with lung parenchymal involvement >25% on CT scan [27]; HIV-seropositive patients; body mass index $\geq$40 kg/m$^2$; and inability to perform CT scan and/or PFTs.

The protocol was approved by the Research Ethics Committee of the Pedro Ernesto University Hospital of the State University of Rio de Janeiro under protocol number CAAE: 87594518.4.0000.52592. Written informed consent and verbal consent prior to enrolment was mandatory. Anonymous personal identifiers were used for each participant.

### Health Assessment Questionnaire-Disability Index (HAQ-DI)

The patients were assessed with the HAQ-DI through a self-administered questionnaire containing 20 questions divided into eight categories on daily activities to assess the functional capacity of the patients. The value of the disability index ranges from 0 to 3 for each question: 0 (no difficulty); 1 (some difficulty); 2 (great difficulty or need for assistance); or 3 (inability

to perform). The final HAQ-DI is the arithmetic mean of the highest scores of the eight categories [28].

## Clinical Disease Activity Index (CDAI)

To assess RA activity, we used the CDAI, which is calculated as the arithmetic sum of four variables: number of painful joints; number of swollen joints; visual analog scale score of global disease activity according to the patient; and visual analog scale score of disease activity according to the physician. The CDAI can be interpreted as remission (CDAI $\leq$ 2.8), low activity (2.8 <CDAI $\leq$10), moderate activity (10 <CDAI$\leq$ 22) or no activity (CDAI >22) [29].

## Laboratory tests

Anti-CCP antibody levels were measured by immunofluorimetric assay and considered positive if the value >10 EliA U/ml. Rheumatoid factor (RF) was measured by the latex agglutination technique and was considered reactive if >8 IU/mL. Antinuclear antibody (ANA) was evaluated by indirect immunofluorescence with HEp-2 cells as substrate, and a titer $\geq$ 1:80 was considered positive. The erythrocyte sedimentation rate (ESR) in the first hour was determined using the Westergren method and was considered normal up to 20 mm. High-sensitivity C-reactive protein (CRP) was evaluated by immunoturbidimetry, and reference values up to 8 mg/L were considered normal.

## Pulmonary function tests

Spirometry, whole-body plethysmography, diffusing capacity of the lungs for carbon monoxide (DLCO) and the $SBN_2W$ test were performed on an HDpft 3000 device (nSpire Health, Inc., Longmont, CO, USA), following the recommendations of the American Thoracic Society/European Respiratory Society [30], using the Brazilian reference values for the evaluated parameters [31–34]. The $SBN_2W$ technique is based on the principle of conservation of mass of an inert gas ($N_2$) present in the lungs, which is progressively washed out with continuous analysis. The test analyzes the concentration of $N_2$ during expiration of the vital capacity at low flow after a single inspiration of 100% $O_2$. The patient first attaches to a mouthpiece and performs ventilatory cycles smoothly. Next, they are asked to exhale up to the residual volume (RV) and subsequently inspire 100% $O_2$ up to the total lung capacity (TLC). Then, exhalation is performed slowly and uniformly at a flow rate of 0.3 to 0.5 L/s up to the RV. The 100% $O_2$ dilutes the $N_2$ present in the lungs until the alveolar $N_2$ concentration is approximately 1%, indicating the end of the test [17]. The exhaled $N_2$ concentration is then measured using a device located at the opening of the airways. We acquired the SIII value, which was analyzed in relation to the predicted value. It is typically represented as %$\Delta N_2$ per liter of lung volume, and the normal range is between 0.5% and 1.5% [17].

## CT acquisition protocol and interpretation

Chest CT examinations were performed at the Pedro Ernesto University Hospital and the D'Or Institute for Research and Teaching. The images were obtained during apnea at maximum inspiration and the end of expiration at a reduced radiation dose. Acquisitions were performed in the craniocaudal direction in the supine position using two different scanners (Brilliance 64, Philips Medical Systems and Revolution CT, General Electric HealthCare). Then, high-resolution reconstruction was performed. Standardized breathing instructions were provided to all patients. The CT acquisition parameters were 120 kVp, automatically variable mAs, pitch of 1, and reconstruction with standard and high-frequency filters (thickness:

1.25–2 mm and interval: 1.25–2 m). No patient received intravenous contrast media. The scans were read by two independent radiologists (FMCR and RSR, with 11 and 25 years of experience in thoracic radiology, respectively), both of whom were unaware of the clinical history or complementary examination results of the patients. The score was given in consensus. The following findings were considered to be nonabnormal on CT: hanging opacities; subpleural curvilinear lines; linear atelectasis; parenchymal bands; small noncalcified nodules smaller than 8 mm; diffusely calcified nodules; pulmonary lymph nodes; normal apical pleural thickening; paraspinal fibrosis secondary to osteophytes; and incidental cysts [35–37]. The following findings were considered abnormal: emphysema; ground-glass opacities; reticular opacities; and nodules [36–38]. All tomographic findings were evaluated using images obtained during inspiration, except for air trapping assessed using images obtained during expiration. To assess the extent of pulmonary abnormalities, the lungs were evaluated according to the five functional lobes. Each lobe was evaluated on a five-point scale according to the percentage of involvement [39]: 0: normal; 1: 1–25%; 2: 26–50%; 3: 51–75% and 4: 76–100%. Then, the sum of the scores of the 5 lobes was calculated to define the extension as follows: 1–5 points: minimum; 6–10 points: mild; 11–15 points: moderate and 16–20: severe [39, 40]. The tomographic patterns were classified into SAD [41] and ILD, which included usual interstitial pneumonia (UIP) and other non-classifiable patterns [42].

## Statistical analysis

The normality of the data distribution was assessed using the Shapiro-Wilk test. The results are expressed as suitable measures of central tendency and dispersion for numerical data (mean ± standard deviation (SD) or median and interquartile ranges (IQRs)) and as frequency and percentage for categorical data. Comparisons between the two groups (patients with or without respiratory symptoms) regarding chest CT scans were analyzed using the chi-square test or Fisher's exact test; comparisons were made when n $\geq$ 5. The relationship of SIII as the percentage of predicted with the numerical variables was analyzed using Spearman correlation analysis and with the categorical variables using the Mann-Whitney U test or Kruskal-Wallis ANOVA. For the purposes of comparative analysis, CT scans were grouped into only two subgroups due to the reduced number of participants for some categories that assess the extent of pulmonary abnormalities. Multivariate analysis was performed using multiple linear regression, which identified the independent variables that explained the variability of the logarithm of SIII (ln SIII), which was used instead of SIII to adapt the distribution to a parametric approach. Variable selection was performed with the stepwise forward method at the 5% level, which selects the smallest subgroup of independent variables that best explains the dependent variable. The following models were constructed: 1) model including only inflammatory markers and pulmonary function parameters (model #1); 2) model including only CT features (model #2); and 3) model including clinical data, inflammatory markers, lung function parameters and CT features (final model). Finally, the performance of the final model was evaluated by analyzing the distribution of residuals using the Shapiro-Wilk test and a graphical evaluation of the histogram. Calibration was verified by graphical analysis of the observed values with the regression line showing the slope and intercept and by plotting the Bland-Altman limits of agreement. Statistical significance was considered if p<0.05. Data analysis was performed using SAS 6.11 software (SAS Institute, Inc., Cary, NC, USA).

## Results

Of the 67 patients who were evaluated for inclusion in the study, 2 were excluded due to a smoking history >10 pack-years prior to the study, and 5 were excluded due to an inability to

perform PFTs, i.e., inappropriate maneuvers in the $SBN_2W$ test (n = 3), spirometry (n = 1) and whole-body plethysmography (n = 1). Among the 60 study participants, the mean age was 53.3 ± 11.3 years, and 55 (91.7%) were women. The median disease duration was 17.5 (9.8–22.7) years, while the time between the onset of joint symptoms and the diagnosis was 14.5 (9–36) months. ANAs were positive in 22 (41.5%) participants. The presence of respiratory symptoms was reported by 32 (53.3%) participants. The presence of comorbidities and extra-articular manifestations was reported by 45 (75%) and 33 (55%) participants, respectively. The medications most reported by the participants were methotrexate (n = 39, 65%), leflunomide (n = 18, 30%) and prednisone (n = 15, 25%). Twelve (20%) participants had a CDAI compatible with high disease activity (>22), while eight (13.3%) had a HAQ-DI compatible with severe functional loss (>2). The median CDAI and HAQ-DI values in the sample were 11 (5–19) and 1 (0.53–1.62), respectively.

Regarding the PFTs, we observed that 11 (18.3%) participants had a forced vital capacity (FVC) <80% of that predicted in the spirometric test, while a forced expiratory volume in 1 second to FVC ratio ($FEV_1$/FVC) <70%, indicative of obstructive damage, was diagnosed in only 3 (5%) participants. On whole-body plethysmography, 15 (25%) participants had a TLC <80% of predicted, confirming restrictive damage; the RV and RV/TLC were >120% of predicted in 3 (5%) and 4 (6.7%) participants, respectively. A total of 19 (31.7%) participants showed a DLCO <80% of the predicted value. In the $SBN_2W$ test, 39 (65%) participants had an SIII >120%. The demographic characteristics, clinical data and pulmonary function results are shown in Table 1.

Regarding CT, we observed that most participants had minimal (n = 26, 43.3%) or mild (n = 14, 23.3%) involvement. The main CT findings were air trapping (n = 36, 64.3%), followed by subpleural ground-glass opacities (n = 13, 23.2%) and subpleural reticulations (n = 12, 21.4%). Traction bronchiectasis and honeycombing were observed only in patients with respiratory symptoms. The results of the participants' CT exams considering the presence or absence of respiratory symptoms are presented in Table 2.

Regarding the univariate correlation analysis, we observed that SIII, as the % of predicted, was significantly correlated with age (r = 0.56, p<0.0001), HAQ-DI (r = 0.34, p = 0.008), FVC (r = -0.67, p<0.0001), TLC (r = -0.46, p = 0.0002), RV/TLC (r = 0.44, p = 0.0004), and DLCO (r = -0.45, p = 0.0003) (Table 3). Higher SIII values were observed in subgroups of participants with the following findings: respiratory symptoms (p = 0.042); moderate/severe disease on CT scans (p = 0.0004); traction bronchiectasis (p = 0.0009); subpleural reticulations (p = 0.003); and honeycombing (p = 0.003) (Fig 1). Regarding bronchiectasis, there was a tendency towards higher SIII values, although without statistical significance (p = 0.063).

In model #1 of multiple linear regression, only two variables were associated with SIII, explaining 54% of its variability: FVC (L) and RV/TLC (%). In model #2, three variables were associated with SIII, explaining 36% of its variability: extension of moderate/severe involvement, subpleural reticulation and bronchiectasis. In the final model, three variables were associated with SIII, explaining 59% of its variability: FVC (L), extent of moderate/severe involvement and age. Table 4 shows the stepwise forward regression analysis for the determination of SIII in our study. Regarding the performance of the regression model, the Shapiro-Wilk test showed that the distribution of nonstandardized residuals was approximately normal, with p = 0.48 (S1 Fig). Regarding the calibration of the regression model, the points were randomly distributed along the adjusted regression, indicating the absence of bias of the differences in relation to the mean (S2 Fig). Finally, the Bland-Altman limits of agreement showed that the vast majority of the differences were within the limits of agreement, with a random distribution along the mean values (S3 Fig).

**Table 1. Demographic characteristics, clinical data and pulmonary function results of patients with rheumatoid arthritis (n = 60).**

| Variable | Value |
|---|---|
| **Demographic data** | |
| Females | 55 (91.7%) |
| Age (years) | 53.3 ± 11.3 |
| **Race** | |
| Caucasian | 28 (46.7%) |
| Non-Caucasian | 25 (41.7%) |
| Black | 7 (11.7%) |
| **MRC dyspnea scale** | |
| 0 | 24 (40%) |
| 1 | 14 (23.3%) |
| 2 | 16 (26.7%) |
| 3 | 6 (10%) |
| **Inflammatory marker** | |
| ESR (mm/h) | 27 (12–45) |
| CRP (mg/dL) | 4.1 (1.6–11.2) |
| **Autoantibodies pattern** | |
| RF (IU/mL) | 64 (9–256) |
| Anti-CPP (IU/ml) | 175 (42–340) |
| **Concomitant drugs for RA** | |
| Methotrexate | 39 (65%) |
| Leflunomide | 18 (30%) |
| Prednisone | 15 (25%) |
| Hydroxychloroquine | 6 (10%) |
| Adalimumab | 5 (8.3%) |
| Others | 10 (16.7%) |
| **Spirometry** | |
| FVC (L) | 2.74 ± 0.74 |
| FVC (% predicted) | 94.2 ± 17.5 |
| $FEV_1$ (L) | 2.23 ± 0.62 |
| $FEV_1$ (% predicted) | 93.6 ± 16.7 |
| $FEV_1$/FVC (%) | 81.9 ± 6.7 |
| **Body plethysmography** | |
| TLC (L) | 4.07 ± 0.95 |
| TLC (% predicted) | 89.1 ± 15.7 |
| RV (L) | 1.40 ± 0.68 |
| RV (% predicted) | 78.7 ± 25.8 |
| RV/TLC (%) | 88.4 ± 21.7 |
| **Pulmonar diffusion** | |
| DLco (ml/min/mm Hg) | 23.29 ± 4.36 |
| DLco (% predicted) | 86.5 ± 22.1 |
| **Single-breath nitrogen washout test** | |
| SIII (L) | 2.82 (1.42–6.35) |
| SIII (% predicted) | 179 (83–339) |

The values shown are means ± SD, median (interquartile ranges) or number (%).

ESR = erythrocyte sedimentation rate; CRP = C-reactive protein; RF = rheumatoid factor; anti-CCP = anti-cyclic citrullinated peptide antibodies; FVC = forced vital capacity; $FEV_1$ = forced expiratory volume in 1 second; TLC = total lung capacity; RV = residual volume; DLco = diffusing capacity for carbon monoxide; SIII = phase III slope of the nitrogen single-breath washout.

**Table 2. Findings observed in chest computed tomography scans of the patients studied considering the presence or absence of respiratory symptoms.**

| Variable | Total sample (n = 60) | Patients with respiratory symptoms (n = 32) | Patients without respiratory symptoms (n = 28) | p-value |
|---|---|---|---|---|
| **Extension of lung involvement** | | | | |
| Normal | 7 (12.5%) | 2 (6.5%) | 5 (20%) | 0.13 |
| Minimum | 26 (46.4%) | 13 (41.9%) | 13 (52%) | |
| Mild | 14 (25%) | 9 (29%) | 5 (20%) | |
| Moderate | 7 (12.5%) | 5 (16.1%) | 2 (8%) | |
| Severe | 2 (3.6%) | 2 (6.5%) | 0 (0%) | |
| **Findings** | | | | |
| Air trapping | 36 (64.3%) | 21 (67.7%) | 15 (60%) | 0.55 |
| Subpleural ground-glass opacities | 13 (23.2%) | 9 (29%) | 4 (16%) | 0.25 |
| Subpleural reticulation | 12 (21.4%) | 9 (29%) | 3 (12%) | 0.12 |
| Traction bronchiectasis | 8 (14.3%) | 8 (25.8%) | 0 (0%) | **0.005** |
| Bronchiectasis | 6 (10.7%) | 5 (16.1%) | 1 (4%) | 0.15 |
| Honeycombing | 6 (10.7%) | 6 (19.4%) | 0 (0%) | **0.022** |
| Emphysema | 6 (10.7%) | 4 (12.9%) | 2 (8%) | 0.44 |
| Opacities/centrilobular nodules | 5 (8.9%) | 4 (12.9%) | 1 (4%) | 0.25 |
| Nodules | 5 (8.9%) | 2 (6.5%) | 3 (12%) | 0.39 |
| Non-subpleural ground-glass opacities | 3 (5.4%) | 2 (6.5%) | 1 (4%) | N/A |
| Diffuse ground-glass opacities | 2 (3.6%) | 2 (6.5%) | 0 (0%) | N/A |
| Consolidation | 2 (3.6%) | 2 (6.5%) | 0 (0%) | N/A |
| Mucoid impaction | 2 (3.6%) | 2 (6.5%) | 0 (0%) | N/A |
| Parietal/bronchial thickening | 2 (3.6%) | 2 (6.5%) | 0 (0%) | N/A |
| Non-subpleural reticulation | 1 (1.8%) | 0 (0%) | 1 (4%) | N/A |

The values shown are number (%). The values in bold refer to significant differences. N/A = Not applicable.

## Discussion

The main findings of the present study were that in patients with RA, almost two-thirds showed maldistribution of ventilation associated with the presence of respiratory symptoms and deterioration of physical function (HAQ-DI). In these patients, a poor distribution of ventilation was related to lower lung volumes, air trapping and damage to pulmonary diffusion. A maldistributed ventilation was associated with the extent of the disease on CT and with certain tomographic patterns, including traction bronchiectasis, subpleural reticulations and honeycombing. Importantly, FVC, disease extent on CT and age together explained 59% of the variability in the distribution of lung ventilation. To our knowledge, this is the first study to evaluate the contributors to the maldistribution of ventilation in patients with RA.

The RV/TLC ratio has important clinical significance in patients with RA because it serves as a functional marker of air trapping. Interestingly, our results showed a direct correlation between SIII and RV/TLC. This indicates that the ventilation distribution should be evaluated in patients with air trapping, which was one of the main tomographic findings in our sample. In line with our study, Lin et al. [43] also observed a high prevalence of air trapping in patients with RA. It is important to note that our findings showed a direct correlation between the SIII and HAQ-DI, suggesting that the maldistribution of ventilation, which in turn causes dyspnea, may influence disease severity (high HAQ-DI). Moreover, ventilation in RA may be affected by musculoskeletal involvement partially related to previous treatment (e.g., steroids) and decreased activity of daily living [23]. A prospective cohort study showed that joint activity in

**Table 3. Spearman's correlation coefficients for phase III slope of the nitrogen single-breath washout, clinical data, inflammatory markers and pulmonary function.**

| Variable | SIII (% predicted) | |
|---|---|---|
| | $r_s$ | p-value |
| **Clinical data** | | |
| Age (years) | 0.56 | **< 0.0001** |
| Disease duration (years) | 0.10 | 0.46 |
| Time between onset of symptoms and diagnosis (months) | 0.08 | 0.56 |
| CDAI (score) | 0.20 | 0.12 |
| HAQ-DI (score) | 0.34 | **0.008** |
| **Inflammatory marker** | | |
| ESR (mm/h) | 0.19 | 0.15 |
| CRP (g/dL) | 0.13 | 0.33 |
| **Autoantibodies pattern** | | |
| RF (IU/mL) | 0.17 | 0.21 |
| Anti-CPP (IU/ml) | 0.06 | 0.69 |
| **Pulmonary function** | | |
| FVC (L) | -0.67 | **< 0.0001** |
| FVC (% predicted) | -0.53 | **< 0.0001** |
| $FEV_1$ (L) | -0.65 | **< 0.0001** |
| $FEV_1$ (% predicted) | -0.51 | **< 0.0001** |
| $FEV_1/FVC$ (%) | -0.17 | 0.21 |
| TLC (L) | -0.46 | **0.0002** |
| TLC (% predicted) | -0.26 | **0.045** |
| RV (L) | 0.12 | 0.35 |
| RV (% predicted) | 0.10 | 0.40 |
| RV/TLC (%) | 0.44 | **0.0004** |
| DLco (ml/min/mm Hg) | -0.40 | **0.0005** |
| DLco (% predicted) | -0.45 | **0.0003** |

The values in bold refer to significant differences.

SIII = phase III slope of the nitrogen single-breath washout; CDAI = Clinical Disease Activity Index; HAQ-DI = Health Assessment Questionnaire-Disability Index; RF = rheumatoid factor; anti-CCP = anti-cyclic citrullinated peptide antibodies; FVC = forced vital capacity; $FEV_1$ = forced expiratory volume in 1 second; TLC = total lung capacity; RV = residual volume; DLco = diffusing capacity for carbon monoxide.

RA was associated with a higher risk of developing ILD [44]. Our results are in agreement with the literature, indicating that changes in ventilation are related to deterioration in physical function; this reinforces the importance of determining the HAQ-DI at each appointment in clinical practice to evaluate the patient at follow-up [26]. The Early Rheumatoid Arthritis Study (ERAS) group previously revealed an association between RA-ILD and a high HAQ-DI [45].

Patients with respiratory symptoms, accounting for more than half of the cohort, had a significantly higher SIII than those without. This finding had already been established in a previous study [22]; however, in other studies, the majority of patients with RA-ILD were asymptomatic [46]. Consequently, they showed that the presence of respiratory symptoms should not be the only variable by which ILD is predicted in patients with RA because joint limitation can be a confounding factor, underestimating the patient's symptoms [46]. It is important to note that the presence of RA-related autoantibodies, especially in high titers, has

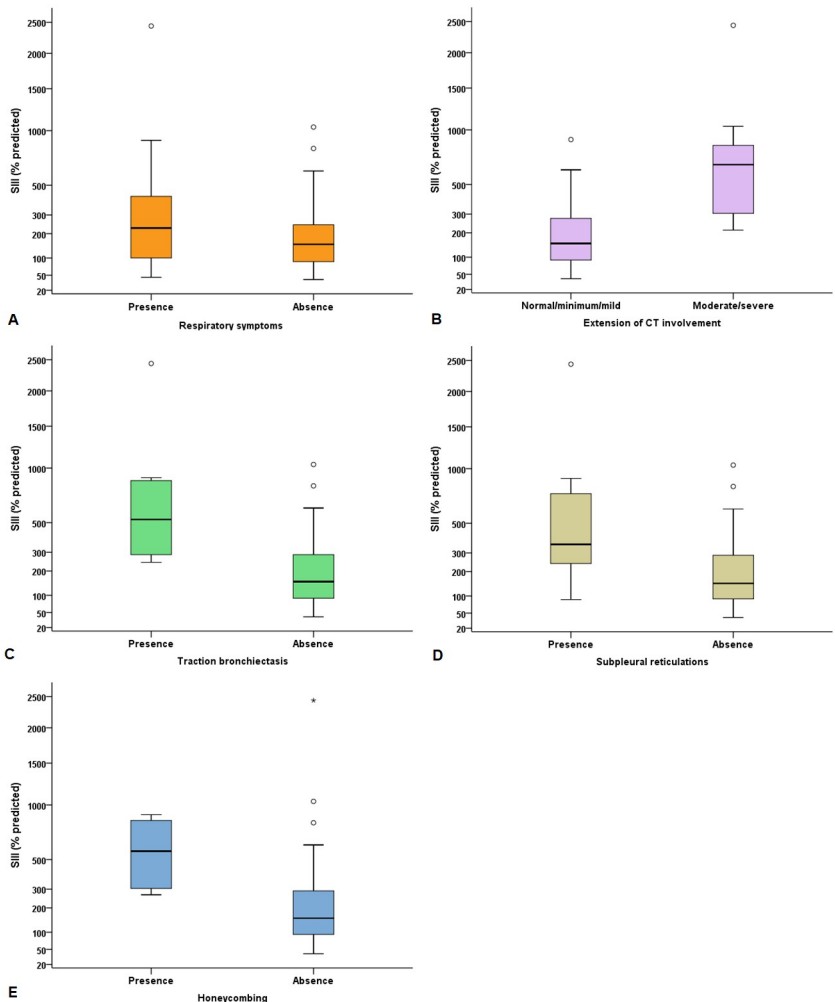

**Fig 1. Phase III slope of the nitrogen single breath washout (SIII) values according to respiratory symptoms (p = 0.042), the extent of involvement on chest computed tomography (CT) (p = 0.0004), traction bronchiectasis (p = 0.0009), subpleural reticulations (p = 0.003) and honeycombing (p = 0.003).**

been suggested as a risk factor for RA-ILD [47, 48], predicting an alteration in pulmonary ventilation and a positive correlation. However, our study showed no correlation between autoantibodies and SIII. A possible explanation for this finding is that almost 60% of our sample had CT scans with normal/minimal involvement. Another possible explanation is that high anti-CCP and/or RF titers are related to smoking, and we excluded patients with a smoking history >10 pack-years [49–51]; indeed, non-coding DNA epigenetic marks suggest that they are possibly induced by environmental factors, with smoking being the most obvious environmental factor for the development of seropositive RA and/or RA-ILD [49, 52]. Furthermore, our populations may be genetically less susceptible to the influence of autoantibodies in the lungs; indeed, a meta-analysis showed that high anti-CCP titers in Asians, Africans and Europeans were significantly related to RA-ILD risk, whereas no association was seen in Americans [47].

The main mechanisms by which ventilation heterogeneity emerges are related to gas transport in the airways close to the terminal bronchioles (convection-dependent), in the peripheral airways (limitation of pulmonary diffusion) and, finally, in the interaction between convection

**Table 4. Stepwise forward regression analysis for the phase III slope of the nitrogen single-breath washout using clinical data, inflammatory markers, pulmonary function parameters and computed tomography features.**

| Variables | B | SEB | p-value | $R^2$ | Adjusted $R^2$ |
|---|---|---|---|---|---|
| **Model #1\*** | | | | | |
| Constant | 6.70 | 0.59 | <0.0001 | - | - |
| FVC (L) | -0.812 | 0.123 | <0.0001 | 0.72 | 0.51 |
| RV/TLC (%) | 0.009 | 0.004 | 0.049 | 0.75 | 0.54 |
| **Model #2†** | | | | | |
| Constant | 4.89 | 0.12 | <0.0001 | - | - |
| Extension of moderate/severe involvement | 1.040 | 0.284 | 0.0005 | 0.53 | 0.26 |
| Subpleural reticulation | 0.619 | 0.255 | 0.018 | 0.58 | 0.31 |
| Bronchiectasis | 0.710 | 0.315 | 0.028 | 0.63 | 0.36 |
| **Final Model‡** | | | | | |
| Constant | 4.87 | 0.92 | <0.0001 | - | - |
| FVC (L) | -0.468 | 0.165 | 0.006 | 0.72 | 0.50 |
| Extension of moderate/severe involvement | 0.771 | 0.266 | 0.005 | 0.74 | 0.53 |
| Age (years) | 0.027 | 0.010 | 0.008 | 0.78 | 0.59 |

\*Model including only inflammatory markers and pulmonary function parameters.

†Model including only computed tomography features.

‡Model including clinical data, inflammatory markers, pulmonary function parameters and computed tomography features.

B = regression coefficient; SEB = standard error of the regression coefficient; $R^2$ = determination coefficient; FVC = forced vital capacity; RV = residual volume; TLC = total lung capacity.

and diffusion [17]. In addition to changes in the small airways, for which it is highly sensitive, the SBN$_2$W test is a promising method for the early detection of any structural abnormality, including in the large airways. Since inflammation in RA can affect almost all components of the lung structure, such as the parenchyma, airways, pleura and pulmonary vasculature, the study of the distribution of ventilation may provide a more sensitive indicator for assessing earlier involvement in these patients because, in the presence of mild pulmonary involvement, spirometry and clinical symptoms do not have good sensitivity. In the present study, which involved a sample among whom almost 75% of the patients had never smoked, there was no significant influence of smoking. This fact is important, as smoking plays a substantial role in the development of RA with positive anti-CCP and in RA-ILD [50, 51]. In addition, smoking plays a role in the pathogenesis of emphysema, and consequently, would bias the changes in the distribution of ventilation in our patients. Another previous finding is that smokers are more likely to have citrullinated proteins in their bronchoalveolar lavage [53], which is related to the development of RA and higher disease activity [54].

In the present study, we observed that almost two-thirds of the patients had maldistribution of ventilation, despite >70% of the patients having minimal/mild pulmonary involvement. The most frequent CT findings were air trapping, subpleural ground-glass opacities, sub-pleural reticulations, traction bronchiectasis/bronchiolectasis and bronchiectasis not related to fibrosis. Data in the literature on the prevalence of RA-ILD are heterogeneous, varying depending on the method and diagnostic criteria used. Studies using CT have revealed that the majority of patients have interstitial abnormalities, followed by airway disease, pleural effusion and rheumatoid nodules [55]. Among the various types of pulmonary involvement in RA, airway disease has been reported as the earliest manifestation [56]. Our imaging findings differ from the literature, as air trapping was seen in more than 60% of the cases. However, we

observed that the evaluation of ventilation distribution was not associated with air trapping but with certain tomographic findings, including traction bronchiectasis, subpleural reticulations and honeycombing. These radiological changes are determinants of fibrosis and are related to changes in gas transport in the conduction zones, close to the terminal bronchioles and in the transition and respiratory zones. It was observed that the subgroup with traction bronchiectasis had significantly higher SIII values than the subgroup without traction bronchiectasis. The subgroup with subpleural reticulations and honeycombing also had significantly higher SIII values than the corresponding negative subgroups, demonstrating the relevance of the findings of fibrosis in the association with SIII. It is also noteworthy that although no single radiological finding served as an independent predictor of SIII, a moderate/severe extent of pulmonary involvement did, indicating that the degree of SIII abnormality is generally proportional to the amount of pulmonary involvement in underlying lung disease, both in relation to ILD and SAD, as a cause of ventilation heterogeneity [57].

In the present study, after adjusting for clinical variables, multivariate analyses revealed that FVC was an independent predictor of SIII. This means that the reduction in lung volume has an important relationship with the heterogeneity of ventilation. Since SIII is measured during a slow expiration of vital capacity, it is not surprising that it correlates more strongly with FVC (L) than with FVC (% predicted). The inclusion of FVC (L) in our final multivariate model possibly allowed age to also enter the stepwise forward selection because lung function is determined by sex, age, height and ancestry [58]. In the sample studied, most patients did not have volume reduction (i.e., FVC and/or TLC <80% of predicted); however, the presence of volume reduction observed in about a quarter of the sample was sufficient to explain the ventilatory distribution heterogeneity in the final multivariate model. In fact, the overall explanatory power of the final model was 59%, with FVC being the main variable explaining the distribution of ventilation. In line with our findings, Gennimata et al. [59] reported an interrelationship between FVC and SIII in COPD patients. The $SBN_2W$ test shows abnormal results for both restrictive and obstructive damage, which suggests its importance in the evaluation of abnormalities in the mechanical properties of the lungs. In both fibrosis and emphysema conditions, the lung is diffusely affected, but the process is not homogeneously distributed. Thus, some regions of the lung fill and empty more slowly than others, leading to abnormal SIII. In addition, we also observed an inverse correlation between SIII and DLCO. This indicates that the greater the damage to the alveolar-capillary membrane, the greater the ventilation heterogeneity is, reinforcing that the degree of abnormality in the $SBN_2W$ test results may indicate the expected difficulties in gas exchange with DLCO reduction [57].

Age was an independent predictor of SIII in our final model, which may be related to the change in the degree of pulmonary impairment due to RA over the years, with progressive deterioration of ventilation. According to Song et al. [60], the progression of the disease in patients with RA and a UIP pattern is variable and depends on age as well as on the functional evaluation at the time of diagnosis and on its serial evolution. Furthermore, as previously shown in some studies, age is a well-documented clinical variable and is considered one of the risk factors for the development of RA-ILD [61]. Importantly, all the variables included in our final model are used in clinical practice, which may help clinicians estimate the ventilation distribution without the need to perform the $SBN_2W$ test, which is expensive and poorly available in most centers. Therefore, our model was able to predict the outcome of ventilation changes in patients with RA, making it an important tool for treatment decision-making and evaluating patient prognosis.

The present study has some limitations that should be considered when interpreting the results. First, the study was conducted in a single center and was cross-sectional in nature,

which precludes the analysis of a cause-effect relationship. Second, we did not have a control group; however, the assessment of pulmonary function values as percentages of predicted values helped minimize the effects of the absence of a control group. Third, we excluded patients with a smoking burden >10 pack-years capable of negatively affecting lung function [62]; however, this fact may prevent the generalization of our results due to the association between RA and smoking. Finally, the interpretation of CT scans was consensual and inter-reader variability was not assessed. Despite these limitations, our multivariate models provide valuable results that explain the heterogeneity in ventilation in patients with RA in routine practice, reinforcing the relevance of the findings, as no other studies to date have been published on this topic. Future studies should assess the applicability of these models, and their reproducibility in other samples should be tested.

In conclusion, FVC, extent of lung involvement and age, variables that are easily obtained in clinical practice, predicted ventilatory maldistribution in patients with RA. In addition, the presence of respiratory symptoms and physical function are closely related to the distribution of ventilation, reinforcing the importance of controlling early and constant systemic inflammation in RA.

## Supporting information

**S1 Fig. Performance of the final regression model with the Shapiro-Wilk test.** The distribution of unstandardized residuals was approximately normal (p = 0.48).
(TIF)

**S2 Fig. Calibration plot of the observed vs predicted values for the phase III slope of the nitrogen single-breath washout (SIII).** There is a narrow slope between the adjusted regression line and the main diagonal, and a strong correlation between the observed and the respective predicted data (r = 0.74; p<0.0001).
(TIF)

**S3 Fig. Limits of agreement plot of the averaged values and the differences (observed—predicted values) for phase III slope of the nitrogen single-breath washout (SIII).** The mean difference was 0.02 with a standard deviation of 0.62, obtaining relatively narrow 95% intervals of agreement (-1.19 for lower and 1.23 for higher).
(TIF)

**S1 Checklist. STROBE statement—Checklist of items that should be included in reports of observational studies.**
(PDF)

## Author Contributions

**Conceptualization:** Elizabeth Jauhar Cardoso Bessa, Felipe de Miranda Carbonieri Ribeiro, Rosana Souza Rodrigues, Cláudia Henrique da Costa, Rogério Rufino, Geraldo da Rocha Castelar Pinheiro, Agnaldo José Lopes.

**Data curation:** Elizabeth Jauhar Cardoso Bessa, Felipe de Miranda Carbonieri Ribeiro, Geraldo da Rocha Castelar Pinheiro, Agnaldo José Lopes.

**Formal analysis:** Felipe de Miranda Carbonieri Ribeiro, Rosana Souza Rodrigues, Geraldo da Rocha Castelar Pinheiro, Agnaldo José Lopes.

**Funding acquisition:** Agnaldo José Lopes.

**Investigation:** Elizabeth Jauhar Cardoso Bessa, Felipe de Miranda Carbonieri Ribeiro, Rosana Souza Rodrigues, Agnaldo José Lopes.

**Methodology:** Elizabeth Jauhar Cardoso Bessa, Felipe de Miranda Carbonieri Ribeiro, Rosana Souza Rodrigues, Cláudia Henrique da Costa, Rogério Rufino, Geraldo da Rocha Castelar Pinheiro, Agnaldo José Lopes.

**Project administration:** Elizabeth Jauhar Cardoso Bessa.

**Supervision:** Agnaldo José Lopes.

**Validation:** Elizabeth Jauhar Cardoso Bessa, Felipe de Miranda Carbonieri Ribeiro, Rosana Souza Rodrigues.

**Writing – original draft:** Elizabeth Jauhar Cardoso Bessa, Felipe de Miranda Carbonieri Ribeiro, Rosana Souza Rodrigues, Cláudia Henrique da Costa, Rogério Rufino, Geraldo da Rocha Castelar Pinheiro, Agnaldo José Lopes.

**Writing – review & editing:** Elizabeth Jauhar Cardoso Bessa, Felipe de Miranda Carbonieri Ribeiro, Rosana Souza Rodrigues, Cláudia Henrique da Costa, Rogério Rufino, Geraldo da Rocha Castelar Pinheiro, Agnaldo José Lopes.

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
