## [Decision Letter · Decision Letter 0]

26 Jul 2023

PONE-D-23-13202Predictive model for ventilatory distribution heterogeneity in patients with rheumatoid arthritisPLOS ONE

Dear Dr. Lopes,

Thank you for submitting your manuscript to PLOS ONE. After careful consideration, we feel that it has merit but does not fully meet PLOS ONE’s publication criteria as it currently stands. Therefore, we invite you to submit a revised version of the manuscript that addresses the points raised during the review process. Please submit your revised manuscript by Sep 09 2023 11:59PM. If you will need more time than this to complete your revisions, please reply to this message or contact the journal office at plosone@plos.org. Please include the following items when submitting your revised manuscript:A rebuttal letter that responds to each point raised by the academic editor and reviewer(s). You should upload this letter as a separate file labeled 'Response to Reviewers'.A marked-up copy of your manuscript that highlights changes made to the original version. You should upload this as a separate file labeled 'Revised Manuscript with Track Changes'.An unmarked version of your revised paper without tracked changes. You should upload this as a separate file labeled 'Manuscript'.

We look forward to receiving your revised manuscript.

Kind regards,

Francesca Pennati, Ph.D.

Academic Editor

PLOS ONE

Journal Requirements:

Reviewers' comments:

Reviewer's Responses to Questions

**Comments to the Author**

1. Is the manuscript technically sound, and do the data support the conclusions?

Reviewer #1: Partly

Reviewer #2: Partly

2. Has the statistical analysis been performed appropriately and rigorously? 

Reviewer #1: Yes

Reviewer #2: Yes

3. Have the authors made all data underlying the findings in their manuscript fully available?

Reviewer #1: Yes

Reviewer #2: Yes

4. Is the manuscript presented in an intelligible fashion and written in standard English?

Reviewer #1: Yes

Reviewer #2: Yes

5. Review Comments to the Author

Reviewer #1: In this paper the Authors investigated the correlation between SIII and data from RA-ILD conventional assessment. The title is informative and the topic very interesting as the availability (and development) of an increasing number of drugs for ILD early treatment. The objective is quite clear as well as the methods. Results support the conclusion and the discussion, even if a bit long-winded, is properly focused on findings.

COMMENTS

- In general a cross sectional-study can not provide data for a predictive model (see L485). Maybe the Authors intended that the objective was to find which of clinical etc etc data are useful in order to identify patients with pathological SIII values.

Abstract

- The numbers of enrolled patients must be moved in Results section

- Conclusion "reinforcing ... in RA": the study findings do not support this statement.

Introduction

- As far I read, the correlation between RA lung structural abnormalities and SIII measure is a mere hypothesis. This "Achille's heel" must be discussed and more data (from other studies) should be provided in the the introduction in order to support the hypothesis.

Methods

- The numbers of enrolled patients must be moved in Results section

- L140 please define what is a "significant" COVID19 lung involvement

- The exclusion of smokers can narrow the "feasibility" of the findings in a general RA population. Please, discuss this issue.

- L149-177 I strongly suggest to summarize or move to supplementary material file

- L221-224 why did you choose the Warrick score instead of the Goh et al. score

Results

- L252 Please specify IQR 9.8-22.7

- Table 1 Drugs: previous or concomitant?

- Consider to merge a Table 2 (L280) with Table 1

- Consider to move Table 3 in supplementary file

- L300-311 report in a Table and or consider to show all these findings in a figure with 5 or 6 box and whiskers plots

- Move Fig3, 5 and 4 in supplementary file

- Fig 2: in order to understand if the sample size is sufficient for the analysis I strongly suggest to show notched box and whiskers plot.

Discussion

- L386-387 consider the alternative hypothesis: dyspnoea could affect HAQ score that it is not related to disease activity (it measures disease severity). Moreover, ventilation can be affected by Musculoskeletal involvement partially related to previous treatment (i.e. steroids) and decrease of daily activity of living

- L404 provide your point of view about this point

- L451 discuss why FVC (L) is related to SIII and not FVC (%) that in clinical practice is widely used as standard in RA-ILD assessment. Moreover, consider that FVC (L) decrease along patients' age.

Reviewer #2: The study on ventilation inhomogeneity in rheumatoid arthritis is interesting and the dataset is of particular relevance. Nevertheless, some major issues are present. First, the authors should state with more accuracy the aim of the study, which is still not well focused in the introduction and in the presentation of the results. Second, from a methodological point of view, the scoring system is not clear to me: if the authors are proposing a novel scoring, they should report the intra- and inter- observer variability. Also, more regression analysis should be performed to reach the goal of the study and to understand which are the structural and functional abnormalities which lead to ventilation heterogeneities.

Introduction.

1) Lines 101-105. “PFTs that measure the distribution of ventilation, on the other hand, are very sensitive to the initial changes that occur in lung structure and function and are therefore useful for the early detection of functional abnormalities, even when other tests are normal, or for confirming airflow obstruction when other tests are only subtly abnormal.” This sentence is not true in general as spirometry for example does not measure the distribution of ventilation and is not sensitive to early abnormalities in lung structure and function. The authors should rephrase or be more specific.

2) Lines 108-110. “developments in the devices responsible for the examination have allowed the SBN2W test to reliably evaluate the inhomogeneity in the distribution of ventilation”. This technique has been proposed to measure the heterogeneity of ventilation. What do the authors intend with “development in the device”?

3) “evaluating the changes in ILD and SAD in RA” Please, rephrase as “evaluating ventilation inhomogeneities in rheumatoid arthritis”

4) Lines 115-117. This paragraph should be moved prior to the introduction of SBN2W as it represents the rationale for using this technique.

5) “phase III slope (SIII)” The authors should specify what this parameter is and if it has been used in other studies.

Methods:

6) line 194: is SIII the only variable measured?

7) Line 200. Please, rephrase the title of the paragraph as “CT acquisition protocol and interpretation”.

8) Lines 207-209. Please provide the reconstruction filter.

9) Lines 213-214. Please, state only that the score was given in consensus.

10) CT interpretation. How the radiologists define what is normal/abnormal?

11) Emphysema was observed on inspiratory or expiratory CT scan?

12) References to the radiology terms should be avoided.

13) Lines 220-221. Was evaluation done on inspiratory or expiratory CT?

14) Is there any reference to the scoring system used by the authors? If the authors introduced a novel radiological scoring system both intra- and inter- reader variability should be evaluated

15) Line 234: Please, correct with pearman correlation analysis.

Results

16) “inability to perform the PFTs” Please specify which PFT test was not performed by these patients.

17) How many patients have respiratory symptoms?

18) Table 3. It would be interesting if the authors report also CT data separately for patients with and without respiratory symptoms, adding 2 columns to the current table.

19) Figure 1 should be removed as the results are already reported in Table 4.

20) Lines 302-304 and Figure 2. The authors should report the results separately for the different classes both in the text and in the figure.

21) Figure 3 should be moved in supplement.

22) Figure 4 and 5 should be reported in one figure with two panels.

23) It would be interesting if the authors could add additional regression analysis, to further investigate which are the main predictors of ventilation inhomogeneity. The first model should include only inflammatory markers and pulmonary function measures. An additional model should include the different CT features to understand which are the main structural alterations which contribute to the functional impairment.

Discussion. In general, I suggest the authors focus the discussion on the comparison with other studies in RA.

24) “This finding reinforces the suitability of the RV/TLC ratio as an indirect marker for assessing SAD”. Please, remove this and remain focused on RA.

25) Lines 387-388. “showing the importance of controlling systemic inflammation to prevent the development of lung ventilation disorders.” The authors should rephrase as: “suggesting that controlling systemic inflammation may prevent the development of lung ventilation disorders” Correlation indicates a relationship between the two variables but not a casual effect of one variable on the other one.

26) “related to lung structure impairment” this is not presented in the results

27) Lines 443-444 “no single radiological finding served as an independent predictor of SIII” The authors should add this analysis to the study as suggested in point 23.

28) Line 451 “indicating the sensitivity of the FVC test”. Please remove this as the sensitivity of FVC to ILD is already well known.

29) Lines 451-453 Please rephrase as the sentence is confusing.

30) Lines 462-463 “the greater the degree of reticular changes and fibrosis”. None of the mentioned parameters are specifically related to reticulations/fibrosis.

31) “In line with our findings, Solomon et al. [57] suggest that RA-ILD patients with greater functional impairment (defined according to the FVC, DLCO or TLC) and those with evidence of disease progression over time (a decline in FVC or 468 DLCO) are at higher risk of death, regardless of the CT image pattern [57].” Please rephrase as the present work does not performed analysis on the risk of death in these patients.

32) Among the limitations I would include that CT interpretation was in consensus and that inter-reader variability was not assessed

6. PLOS authors have the option to publish the peer review history of their article (what does this mean?). If published, this will include your full peer review and any attached files.

Reviewer #1: No

Reviewer #2: No

---

## [Author Response · Author response to Decision Letter 0]

12 Aug 2023

REVIEWER #1:

First, we would like to thank you for your time and comments, which have indeed helped us improve the manuscript. We agree that some points in the initial version of the manuscript should have been described in more detail. We have replied to each of your comments below. The modifications made to the text are highlighted.

In this paper the Authors investigated the correlation between SIII and data from RA-ILD conventional assessment. The title is informative and the topic very interesting as the availability (and development) of an increasing number of drugs for ILD early treatment. The objective is quite clear as well as the methods. Results support the conclusion and the discussion, even if a bit long-winded, is properly focused on findings.

AUTHORS:

Thank you for your comments.

COMMENTS

- In general a cross sectional-study can not provide data for a predictive model (see L485). Maybe the Authors intended that the objective was to find which of clinical etc etc data are useful in order to identify patients with pathological SIII values.

AUTHORS:

The authors completely agree with you. Accordingly, we have made modifications to the title and purpose of the manuscript as follows:

• Title: “Association between clinical, serological, functional and radiological findings and ventilatory distribution heterogeneity in patients with rheumatoid arthritis.”

• Objective: “Thus, the objective of the present study was to find out which clinical, serological, functional and radiological findings are useful to identify RA patients with pathological values of SIII measured by the SBN2W test.”

Abstract

- The numbers of enrolled patients must be moved in Results section.

AUTHORS:

The modification was made accordingly.

- Conclusion "reinforcing ... in RA": the study findings do not support this statement.

AUTHORS:

The modification was made accordingly.

Introduction

- As far I read, the correlation between RA lung structural abnormalities and SIII measure is a mere hypothesis. This "Achille's heel" must be discussed and more data (from other studies) should be provided in the the introduction in order to support the hypothesis.

AUTHORS:

The authors fully agree with their comments. Accordingly, the Introduction section has been extensively reworked. More data from other studies were provided to support the hypothesis. Following their suggestions, the hypothesis was rewritten and the objective was changed accordingly.

Methods

- The numbers of enrolled patients must be moved in Results section.

AUTHORS:

The modification was made accordingly.

- L140 please define what is a "significant" COVID19 lung involvement.

AUTHORS:

As requested, the sentence has been written as follows: “history of COVID-19 pneumonia with lung parenchymal involvement >25% on CT scan [27].”

- The exclusion of smokers can narrow the "feasibility" of the findings in a general RA population. Please, discuss this issue.

AUTHORS:

The authors fully agree with you. In addition to an extensive discussion on the association between RA and smoking (already existing in the previous version of the manuscript), we added this point as a limitation of the study as follows: “Third, we excluded patients with a smoking burden >10 pack-years capable of negatively affecting lung function [62]; however, this fact may prevent the generalization of our results due to the association between RA and smoking.”

- L149-177 I strongly suggest to summarize or move to supplementary material file.

AUTHORS:

As suggested, we have summarized the information contained in L149-177.

- L221-224 why did you choose the Warrick score instead of the Goh et al. score.

AUTHORS:

We sought a more comprehensive score that did not only assess interstitial involvement, as we found many patients with radiological signs of small airway disease.

Results

- L252 Please specify IQR 9.8-22.7.

AUTHORS:

Since the measures of central tendency and dispersion for numerical data are shown throughout the manuscript, we chose to highlight the information in the Statistical analysis section as follows: “The results are expressed as suitable measures of central tendency and dispersion for numerical data (mean ± standard deviation (SD) or median and interquartile ranges (IQRs)) and as frequency and percentage for categorical data.”

- Table 1 Drugs: previous or concomitant?

AUTHORS:

Information has been added accordingly.

- Consider to merge a Table 2 (L280) with Table 1.

AUTHORS:

As suggested, we have emerged Table 2 with Table 1.

- Consider to move Table 3 in supplementary file.

AUTHORS:

As required by Reviewer #2, we provide CT data separately for patients with and without respiratory symptoms in Table 3, adding 2 columns to the current table.

- L300-311 report in a Table and or consider to show all these findings in a figure with 5 or 6 box and whiskers plots.

AUTHORS:

Thank you for your suggestion. All these findings were shown in a figure in the new version of the manuscript. However, the statistical software we used did not allow us to use whisker plots.

- Move Fig 3, 5 and 4 in supplementary file.

AUTHORS:

As required, Fig. 3, 4 and 5 have been moved to supplementary file.

- Fig 2: in order to understand if the sample size is sufficient for the analysis I strongly suggest to show notched box and whiskers plot.

AUTHORS:

As pointed out above, the statistical software we used did not allow us to show the notched box and whiskers plot.

Discussion

- L386-387 consider the alternative hypothesis: dyspnoea could affect HAQ score that it is not related to disease activity (it measures disease severity). Moreover, ventilation can be affected by Musculoskeletal involvement partially related to previous treatment (i.e. steroids) and decrease of daily activity of living.

AUTHORS:

Thank you for your suggestion. The sentences were rewritten as follows: “It is important to note that our findings showed a direct correlation between the SIII and HAQ-DI, suggesting that maldistribution of ventilation, which in turn causes dyspnea, may influence disease severity (high HAQ-DI). Moreover, ventilation in RA may be affected by musculoskeletal involvement partially related to previous treatment (e.g., steroids) and decreased activity of daily living [23].”

- L404 provide your point of view about this point.

AUTHORS:

As required, we added two hypotheses about the lack of correlation between autoantibodies and RA-ILD observed in our study, as follows: “A possible explanation for this finding is that almost 60% of our sample had CT scans with normal/minimal involvement. Another possible explanation is that high anti-CCP and/or RF titers are related to smoking, and we excluded patients with a smoking history >10 pack-years [49-51]; indeed, non-coding DNA epigenetic marks suggest that they are possibly induced by environmental factors, with smoking being the most obvious environmental factor for the development of seropositive RA and/or RA-ILD [49,52]. Furthermore, our populations may be genetically less susceptible to the influence of autoantibodies in the lungs; indeed, a meta-analysis showed that high anti-CCP titers in Asians, Africans and Europeans were significantly related to RA-ILD risk, whereas no association was seen in Americans [47].”

- L451 discuss why FVC (L) is related to SIII and not FVC (%) that in clinical practice is widely used as standard in RA-ILD assessment. Moreover, consider that FVC (L) decrease along patients' age.

AUTHORS:

Thank you for your observations. In our multivariate model, both FVC (L, rs=-0.67, p<0.0001) and FVC (% predicted, rs=-0.53, p<0.0001) were correlated with SIII. We add the explanation for entering FVC (L) and age in the stepwise forward linear regression analysis as follows:

“Since SIII is measured during a slow expiration of vital capacity, it is not surprising that it correlates more strongly with FVC (L) than with FVC (% predicted). The inclusion of FVC (L) in our final multivariate model possibly allowed age to also enter the stepwise forward selection because lung function is determined by sex, age, height and ancestry [58].”

[58] Stanojevic S, Kaminsky DA, Miller MR, Thompson B, Aliverti A, Barjaktarevic I, et al. ERS/ATS technical standard on interpretive strategies for routine lung function tests. The European Respiratory Journal. 2022; 60(1): 2101499.

REVIEWER #2:

First, we would like to thank you for your time and comments, which have indeed helped us improve the manuscript. We agree that some points in the initial version of the manuscript should have been described in more detail. We have replied to each of your comments below. The modifications made to the text are highlighted.

The study on ventilation inhomogeneity in rheumatoid arthritis is interesting and the dataset is of particular relevance. Nevertheless, some major issues are present. First, the authors should state with more accuracy the aim of the study, which is still not well focused in the introduction and in the presentation of the results. Second, from a methodological point of view, the scoring system is not clear to me: if the authors are proposing a novel scoring, they should report the intra- and inter- observer variability. Also, more regression analysis should be performed to reach the goal of the study and to understand which are the structural and functional abnormalities which lead to ventilation heterogeneities.

AUTHORS:

Thank you for your comments. We think that some considerations are important as follows:

1) The objective of the study was stated more precisely following the suggestion given by Reviewer #1. Furthermore, the Introduction section has been extensively reworked to support our hypothesis.

2) The study does not propose a new scoring system. Indeed, we sought to find out which clinical, serological, functional and radiological findings are useful to identify RA patients with pathological values of SIII measured by the SBN2W test. The multivariate model proposed to explain the SIII values was satisfactory. However, we recognize that a future step would be to verify its applicability and, for that, the reproducibility in other samples should be tested. Accordingly, we have added the following limitation in the Discussion section: “Future studies should assess the applicability of these models, and their reproducibility in other samples should be tested.”

3) As requested by you, further regression analyzes were performed to achieve the goal of the study and to understand which are the structural and functional abnormalities that lead to ventilation heterogeneities. Information about these analyzes has been added in Table 4 and throughout the manuscript.

Introduction.

1) Lines 101-105. “PFTs that measure the distribution of ventilation, on the other hand, are very sensitive to the initial changes that occur in lung structure and function and are therefore useful for the early detection of functional abnormalities, even when other tests are normal, or for confirming airflow obstruction when other tests are only subtly abnormal.” This sentence is not true in general as spirometry for example does not measure the distribution of ventilation and is not sensitive to early abnormalities in lung structure and function. The authors should rephrase or be more specific.

AUTHORS:

Thank you for your suggestion. The sentence has been reformulated as follows: “Techniques for assessing small airway function and ventilation distribution—such as the single-breath nitrogen washout test (SBN2W) and impulse oscillometry—, on the other hand, are very sensitive to the initial changes that occur in lung structure and function and are therefore useful for the early detection of functional abnormalities, even when other tests are normal, or for confirming airflow obstruction when other tests are only subtly abnormal [14–16].”

2) Lines 108-110. “developments in the devices responsible for the examination have allowed the SBN2W test to reliably evaluate the inhomogeneity in the distribution of ventilation”. This technique has been proposed to measure the heterogeneity of ventilation. What do the authors intend with “development in the device”?

AUTHORS:

Thank you for your observation. In fact, there have been important advances in recent years both in relation to the standardization of the technique (including the publication of new guidelines) and in the development of devices by manufacturers. Along these lines, the sentences have been reformulated as follows: “In recent years, the development of robust accurate commercial devices and the standardization of quality control of washout systems allowed the inert gas washout test to reliably evaluate the ventilatory distribution inhomogeneity [17,18]. At the same time, there has been increasing application of the SBN2W test in a wide variety of clinical conditions, including SAD, asthma, chronic obstructive pulmonary disease (COPD) and systemic sclerosis [15,16-19–21].”

3) “evaluating the changes in ILD and SAD in RA” Please, rephrase as “evaluating ventilation inhomogeneities in rheumatoid arthritis”.

AUTHORS:

As required, the modification was made accordingly.

4) Lines 115-117. This paragraph should be moved prior to the introduction of SBN2W as it represents the rationale for using this technique.

AUTHORS:

As required, the paragraph was moved prior to the introduction of the SBN2W test.

5) “phase III slope (SIII)” The authors should specify what this parameter is and if it has been used in other studies.

AUTHORS:

Thank you for your suggestion. We have added the following sentences in the Introduction section:

• “The phase III slope (SIII) is the index derived from the SBN2W test that represents alveolar gas; specifically, the SIII is the change in N2 concentration between 25-75% of the expired volume during the SBN2W test. Although the SBN2W test is poorly accessible and is expensive and difficult to perform, it is highly accurate in detecting ventilation maldistribution [21].”

• “Several correlations have been identified between SIII and parameters for monitoring lung diseases. In patients with COPD, SIII was associated with dyspnea, exercise-induced desaturation, lung mechanics and the degree of emphysema [20,25]. In asthma, ventilation heterogeneity assessed by SIII represents an important indicator of poor control and a high exacerbation rate [15]. In SSc patients, SIII was associated with restrictive damage, changes in pulmonary diffusion and CT patterns [21]. In RA, SIII was associated with RF positivity, bronchial involvement on CT and exercise functional capacity [22–24].”

Methods:

6) line 194: is SIII the only variable measured?

AUTHORS:

Multiple breath and single-breath inert gas washout tests (MBW and SBW, respectively) assess the efficiency of ventilation distribution. The inhomogeneity of ventilation distribution is reflected in delayed SBW marker gas clearance (raised SBW phase III slope-SIII) and magnitude and progression of MBW concentration normalized phase III slopes (SnIII) through subsequent breaths [1]. Since we used single-breath nitrogen washout test (SBN2W), then the inhomogeneity in ventilatory distribution was assessed by SIII.

[1] Robinson PD, Latzin P, Verbanck S, Hall GL, Horsley A, Gappa M, et al. Consensus statement for inert gas washout measurement using multiple- and single- breath tests. The European Respiratory Journal. 2013; 41(3):507–22.

7) Line 200. Please, rephrase the title of the paragraph as “CT acquisition protocol and interpretation”.

AUTHORS:

As required, the modification was made accordingly.

8) Lines 207-209. Please provide the reconstruction filter.

AUTHORS:

As required, information has been provided as follows: “The CT acquisition parameters were 120 kVp, automatically variable mAs, pitch of 1, and reconstruction with standard and high-frequency filters (thickness: 1.25-2 mm and interval: 1.25-2 m).”

9) Lines 213-214. Please, state only that the score was given in consensus.

AUTHORS:

As required, the modification was made accordingly.

10) CT interpretation. How the radiologists define what is normal/abnormal?

AUTHORS:

Thank you for your observation. Based on several literature studies cited in the paragraph, findings were defined as normal/abnormal.

The following findings were defined as normal as follows:

• Hanging opacities; subpleural curvilinear lines; linear atelectasis; parenchymal bands; small noncalcified nodules smaller than 8 mm; diffusely calcified nodules; pulmonary lymph nodes:

[1] Hochhegger B, Marchiori E, Rodrigues R, Mançano A, Jasinowodolinski D, Chate RC, et al. Consensus statement on thoracic radiology terminology in Portuguese used in Brazil and in Portugal. Jornal Brasileiro de Pneumologia. 2021; 47(5):e20200595.

[2] Hansell DM, Bankier AA, MacMahon H, McLoud TC, Müller NL, Remy J. Fleischner Society: glossary of terms for thoracic imaging. Radiology. 2008; 246(3):697–722.

[3] MacMahon H, Naidich DP, Goo JM, Lee KS, Leung ANC, Mayo JR, et al. Guidelines for management of incidental pulmonary nodules detected on CT images: from the Fleischner Society. Radiology. 2017; 284(1):228–43.

• Normal apical pleural thickening:

[1] McLoud TC, Isler RJ, Novelline RA, Putman CE, Simeone J, Stark P. The Apical Cap. American Journal of Roentgenology. 1981; 137(2):299–306.

• Paraspinal fibrosis secondary to osteophytes:

[1] Hatabu H, Hunninghake GM, Richeldi L, Brown KK, Wells AU, Remy-Jardin M, et al. Interstitial lung abnormalities detected incidentally on CT: a Position Paper from the Fleischner Society. The Lancet Respiratory Medicine. 2020; 8(7):726–37.

[2] Hata A, Schiebler ML, Lynch DA, Hatabu H. Interstitial lung abnormalities: state of the art. Radiology. 2021; 301(1):19–34.

• Incidental cysts:

[1] Raoof S, Bondalapati P, Vydyula R, Ryu JH, Gupta N, Raoof S, et al. Cystic lung diseases: algorithmic approach. Chest. 2016; 150(4):945–65.

The following findings were defined as anormal as follows:

• Emphysema:

[1] Lynch DA, Austin JHM, Hogg JC, Grenier PA, Kauczor HU, Bankier AA, et al. CT-definable subtypes of chronic obstructive pulmonary disease: a statement of the Fleischner Society. Radiology. 2015; 277(1):192–205.

• Ground-glass opacities; reticular opacities:

[1] Hatabu H, Hunninghake GM, Richeldi L, Brown KK, Wells AU, Remy-Jardin M, et al. Interstitial lung abnormalities detected incidentally on CT: a Position Paper from the Fleischner Society. The Lancet Respiratory Medicine. 2020; 8(7):726–37.

[2] Hata A, Schiebler ML, Lynch DA, Hatabu H. Interstitial lung abnormalities: state of the art. Radiology. 2021; 301(1):19–34.

• Nodules:

[1] MacMahon H, Naidich DP, Goo JM, Lee KS, Leung ANC, Mayo JR, et al. Guidelines for management of incidental pulmonary nodules detected on CT images: from the Fleischner Society. Radiology. 2017; 284(1):228–43.

11) Emphysema was observed on inspiratory or expiratory CT scan?

AUTHORS:

Emphysema was assessed on inspiration images. This information was added to the manuscript as follows: “All tomographic findings were evaluated using images obtained during inspiration, except for air trapping assessed using images obtained during expiration.”

12) References to the radiology terms should be avoided.

AUTHORS:

As required, we have left only the most important references on radiology terms.

13) Lines 220-221. Was evaluation done on inspiratory or expiratory CT?

AUTHORS:

As required, information was added to the manuscript as follows: “All tomographic findings were evaluated using images obtained during inspiration, except for air trapping assessed using images obtained during expiration.”

14) Is there any reference to the scoring system used by the authors? If the authors introduced a novel radiological scoring system both intra- and inter- reader variability should be evaluated.

AUTHORS:

Thank you for your observation. We did not introduce any new radiological scoring system and therefore intra- and inter-reader variability was not assessed. Reference to the scoring system used in our study has been added accordingly [1].

[1] Park WH, Kim SS, Shim SC, Song ST, Jung SS, Kim JH, et al. Visual assessment of chest computed tomography findings in anticyclic citrullinated peptide antibody positive rheumatoid arthritis: is it associated with airway abnormalities? Lung. 2016; 194(1):97–105.

15) Line 234: Please, correct with Spearman correlation analysis.

AUTHORS:

As requested by you, correction has been made accordingly.

Results

16) “inability to perform the PFTs” Please specify which PFT test was not performed by these patients.

AUTHORS:

As required, information has been added in the Results section as follows: “Of the 67 patients who were evaluated for inclusion in the study, 2 were excluded due to a smoking history >10 pack-years prior to the study, and 5 were excluded due to an inability to perform PFTs, i.e., inappropriate maneuvers in the SBN2W test (n=3), spirometry (n=1) and whole-body plethysmography (n=1).”

17) How many patients have respiratory symptoms?

AUTHORS:

As required, we have added information in the Results section as follows: “The presence of respiratory symptoms was reported by 32 (53.3%) participants.”

18) Table 3. It would be interesting if the authors report also CT data separately for patients with and without respiratory symptoms, adding 2 columns to the current table.

AUTHORS:

As requested by you, we provide CT data separately for patients with and without respiratory symptoms in Table 3, adding 2 columns to the current table. In view of this new statistical analysis, the following information was added to the manuscript:

• Statistical analysis: “Comparisons between the two groups (patients with or without respiratory symptoms) regarding chest CT scans were analyzed using the chi-square test or Fisher's exact test; comparisons were made when n ≥ 5.”

• Results: “Traction bronchiectasis and honeycombing were observed only in patients with respiratory symptoms. The results of the participants' CT exams considering the presence or absence of respiratory symptoms are presented in Table 2.”

19) Figure 1 should be removed as the results are already reported in Table 4.

AUTHORS:

As requested, Figure 1 has been removed.

20) Lines 302-304 and Figure 2. The authors should report the results separately for the different classes both in the text and in the figure.

AUTHORS:

Thank you for your observation. The results of the 5 categories separately are presented in Table 2. However, the CT scans were grouped into only two subgroups for the purposes of comparative analysis due to the reduced number of participants for some categories that assess the extent of lung abnormalities. We added this information in the Statistical Analysis section in the new version of the manuscript.

21) Figure 3 should be moved in supplement.

AUTHORS:

As required by you, Fig. 3 have been moved to supplementary file.

22) Figure 4 and 5 should be reported in one figure with two panels.

AUTHORS:

As required by Reviewer #1, Figs. 4 and 5 have also been moved to the supplemental file.

23) It would be interesting if the authors could add additional regression analysis, to further investigate which are the main predictors of ventilation inhomogeneity. The first model should include only inflammatory markers and pulmonary function measures. An additional model should include the different CT features to understand which are the main structural alterations which contribute to the functional impairment.

AUTHORS:

Thanks for your suggestion. We built two new models following your instructions. The regression analysis models are shown in Table 4. In addition, the following information was added in the manuscript about these models:

• Statistical analysis: “The following models were constructed: 1) model including only inflammatory markers and pulmonary function parameters (model #1); 2) model including only CT features (model #2); and 3) model including clinical data, inflammatory markers, lung function parameters and CT features (final model).”

• Results: “In model #1 of multiple linear regression, only two variables were associated with SIII, explaining 54% of its variability: FVC (L) and RV/TLC (%). In model #2, three variables were associated with SIII, explaining 36% of its variability: extension of moderate/severe involvement, subpleural reticulation and bronchiectasis. In the final model, three variables were associated with SIII, explaining 59% of its variability: FVC (L), extent of moderate/severe involvement and age. Table 4 shows the stepwise forward regression analysis for the determination of SIII in our study.”

Discussion.

In general, I suggest the authors focus the discussion on the comparison with other studies in RA.

AUTHORS:

Thank you for your suggestion. The Discussion section has been extensively revised to focus on comparison with other studies in RA.

24) “This finding reinforces the suitability of the RV/TLC ratio as an indirect marker for assessing SAD”. Please, remove this and remain focused on RA.

AUTHORS:

As requested, the removal was made and the sentence was rewritten as follows: “In line with our study, Lin et al. [43] also observed a high prevalence of air trapping in patients with RA.”

25) Lines 387-388. “showing the importance of controlling systemic inflammation to prevent the development of lung ventilation disorders.” The authors should rephrase as: “suggesting that controlling systemic inflammation may prevent the development of lung ventilation disorders” Correlation indicates a relationship between the two variables but not a casual effect of one variable on the other one.

AUTHORS:

Thank you for your observation pointing out the fact that correlation only indicates relationship between the two variables. As suggested by Reviewer #1, the sentences were rewritten as follows: “It is important to note that our findings showed a direct correlation between the SIII and HAQ-DI, suggesting that the maldistribution of ventilation, which in turn causes dyspnea, may influence disease severity (high HAQ-DI). Moreover, ventilation in RA may be affected by musculoskeletal involvement partially related to previous treatment (e.g., steroids) and decreased activity of daily living [23].”

26) “related to lung structure impairment” this is not presented in the results.

AUTHORS:

Thank you for your observation. The sentence was rewritten as follows: “Our results are in agreement with the literature, indicating that changes in ventilation are related to deterioration in physical function; this reinforces the importance of determining the HAQ-DI at each appointment in clinical practice to evaluate the patient at follow-up [26].”

27) Lines 443-444 “no single radiological finding served as an independent predictor of SIII” The authors should add this analysis to the study as suggested in point 23.

AUTHORS:

As requested, additional regression analyzes were performed accordingly.

28) Line 451 “indicating the sensitivity of the FVC test”. Please remove this as the sensitivity of FVC to ILD is already well known.

AUTHORS:

As required, the removal of part of the sentence was made accordingly.

29) Lines 451-453 Please rephrase as the sentence is confusing.

AUTHORS:

Thank you for your observation. The sentence has been rephrased as follows: “In the sample studied, most patients did not have volume reduction (i.e., FVC and/or TLC <80% of predicted); however, the presence of volume reduction observed in about a quarter of the sample was sufficient to explain the ventilatory distribution heterogeneity in the final multivariate model.”

30) Lines 462-463 “the greater the degree of reticular changes and fibrosis”. None of the mentioned parameters are specifically related to reticulations/fibrosis.

AUTHORS:

Thank you for your observation. The sentence has been rephrased as follows: “In addition, we also observed an inverse correlation between SIII and DLCO. This indicates that the greater the damage to the alveolar-capillary membrane, the greater the ventilation heterogeneity is, reinforcing that the degree of abnormality in the SBN2W test results may indicate the expected difficulties in gas exchange with DLCO reduction [57].”

31) “In line with our findings, Solomon et al. [57] suggest that RA-ILD patients with greater functional impairment (defined according to the FVC, DLCO or TLC) and those with evidence of disease progression over time (a decline in FVC or 468 DLCO) are at higher risk of death, regardless of the CT image pattern [57].” Please rephrase as the present work does not performed analysis on the risk of death in these patients.

AUTHORS:

In line with your concern that we should focus the discussion on comparing our results with other RA studies, we excluded the citation from the study by Solomon et al. because the risk of death analysis was not performed in our sample.

32) Among the limitations I would include that CT interpretation was in consensus and that inter-reader variability was not assessed.

AUTHORS:

As required, we have included this limitation in the Discussion section as follows: “Finally, the interpretation of CT scans was consensual and inter-reader variability was not assessed.”

---

## [Decision Letter · Decision Letter 1]

4 Sep 2023

Association between clinical, serological, functional and radiological findings and ventilatory distribution heterogeneity in patients with rheumatoid arthritis

PONE-D-23-13202R1

Dear Dr. LOPES - FAPERJ - CPF: 048267207-26,

We’re pleased to inform you that your manuscript has been judged scientifically suitable for publication and will be formally accepted for publication once it meets all outstanding technical requirements.

Kind regards,

Francesca Pennati, Ph.D.

Academic Editor

PLOS ONE

Additional Editor Comments (optional):

Reviewers' comments:

Reviewer's Responses to Questions

**Comments to the Author**

1. If the authors have adequately addressed your comments raised in a previous round of review and you feel that this manuscript is now acceptable for publication, you may indicate that here to bypass the “Comments to the Author” section, enter your conflict of interest statement in the “Confidential to Editor” section, and submit your "Accept" recommendation.

Reviewer #1: All comments have been addressed

Reviewer #2: All comments have been addressed

2. Is the manuscript technically sound, and do the data support the conclusions?

Reviewer #1: Yes

Reviewer #2: Yes

3. Has the statistical analysis been performed appropriately and rigorously? 

Reviewer #1: Yes

Reviewer #2: Yes

4. Have the authors made all data underlying the findings in their manuscript fully available?

Reviewer #1: Yes

Reviewer #2: Yes

5. Is the manuscript presented in an intelligible fashion and written in standard English?

Reviewer #1: Yes

Reviewer #2: Yes

6. Review Comments to the Author

Reviewer #1: all comments were addressed

Reviewer #2: All my comments were addressed.

7. PLOS authors have the option to publish the peer review history of their article (what does this mean?). If published, this will include your full peer review and any attached files.

Reviewer #1: **Yes: **Alarico Ariani

Reviewer #2: No

---

## [Editor Report · Acceptance letter]

12 Oct 2023

PONE-D-23-13202R1 

Association between clinical, serological, functional and radiological findings and ventilatory distribution heterogeneity in patients with rheumatoid arthritis 

Dear Dr. Lopes:

I'm pleased to inform you that your manuscript has been deemed suitable for publication in PLOS ONE. Congratulations! Your manuscript is now with our production department. 

Kind regards, 

on behalf of

Dr. Francesca Pennati 

Academic Editor

PLOS ONE